# The Effects of *Lactobacillus plantarum* and *Lactobacillus buchneri* on the Fermentation Quality, In Vitro Digestibility, and Aerobic Stability of *Silphium perfoliatum* L. Silage

**DOI:** 10.3390/ani14152279

**Published:** 2024-08-05

**Authors:** Yitong Jin, Peng Wang, Fuhou Li, Meng Yu, Jiarui Du, Tianyue Zhao, Qixuan Yi, Hongyu Tang, Bao Yuan

**Affiliations:** College of Animal Sciences, Jilin University, Changchun 130062, China; ytjin23@mails.jlu.edu.cn (Y.J.); lifh@jlu.edu.cn (F.L.); yumeng22@mails.jlu.edu.cn (M.Y.); dujr23@mails.jlu.edu.cn (J.D.); tianyue22@mails.jlu.edu.cn (T.Z.); yiqx21@mails.jlu.edu.cn (Q.Y.); tanghy@jlu.edu.cn (H.T.)

**Keywords:** aerobic stability, fermentation quality, in vitro digestibility, *Lactobacillus plantarum*, *Lactobacillus buchneri*, *Silphium perfoliatum* L.

## Abstract

**Simple Summary:**

Under the shortage of roughage resources in China, *Silphium perfoliatum* L. (SP) can be an ideal roughage choice for ruminants and has good application prospects. Ensiling is an efficient forage treatment technology for retaining the nutrients and extending the shelf life of forage. At present, there have been no reports on the effects of *Lactobacillus plantarum* and *Lactobacillus buchneri*, whether alone or in combination, on *Silphium perfoliatum* L. silage. Therefore, in this experiment, *L. plantarum* and *L. buchneri* were added individually or combined to SP-silage to investigate the effects of different fermentation types of *Lactobacilli* on the fermentation quality, in vitro digestibility, and aerobic stability of SP-silage. It was found that a mixture of *L. plantarum* and *L. buchneri* provided the best silage results. *L. buchneri* can be used as an additive to improve the aerobic stability of SP-silage. The results of our research provide a certain scientific basis and technical support for obtaining high-quality SP-silage in production.

**Abstract:**

In this experiment, *Lactobacillus plantarum* and *Lactobacillus buchneri* were added individually or in combination to *Silphium perfoliatum* L. (SP) silage to investigate the effects of different fermentation types of lactobacilli on the fermentation quality, in vitro digestibility, and aerobic stability of SP-silage, with a view to providing a certain scientific basis and technical support for obtaining high-quality SP-silage in production. The experiment comprised a non-additive group (control), an *L. plantarum* group (LP), an *L. buchneri* group (LB), and an *L. plantarum* and *L. buchneri* mixed treatment group (LPLB). Samples were taken after 60 days of fermentation and analyzed for the fermentation quality, in vitro digestibility, and aerobic stability of the SP-silage. The results showed that the addition of LP, LB, and LPLB significantly reduced the pH and proportion of ammonia nitrogen to total nitrogen and significantly increased the lactic acid, in vitro dry matter digestibility, and in vitro crude protein digestibility in the SP-silage (*p* < 0.05). Compared to the control group, the dry matter and crude protein contents of the LB and LPLB groups were significantly increased, while the neutral detergent fiber and acid detergent fiber contents were significantly reduced (*p* < 0.05). The SP-silage supplemented with LPLB had the highest dry matter and crude protein contents. The gross and digestible energies of the SP-silage in the LB and LPLB groups were significantly higher than those in the control and LP groups (*p* < 0.05). The aerobic stability of the SP-silage was significantly reduced by 24.14% in the LP group and increased by 58.62% and 34.48% in the LB and LPLB groups, respectively, compared to the control group (*p* < 0.05). It was shown that adding a combination of LP and LB resulted in the best fermentation quality, nutritional value, and in vitro digestibility of the SP-silage. LB was effective in improving the aerobic stability of SP-silage.

## 1. Introduction

As one of the key industries indispensable to a country’s economy and its people’s livelihood, animal husbandry plays a crucial role in maintaining food security, promoting sustainable growth of the rural economy, and facilitating the increase in herdsmen’s incomes. However, in recent years, with the rapid development of China’s livestock industry, the forage supply problem has become increasingly prominent, and the existing forage resources have made it difficult to support the demand for high-quality development of the livestock industry. Therefore, exploring and broadening new feed resources has become an urgent problem in the current development of animal husbandry. *Silphium perfoliatum* L. (SP) is a perennial dicotyledonous herb belonging to the genus Pine Sedge in the family Asteraceae, which is widely adapted, resilient, of good quality, and easy to cultivate, making it a high-quality and high-yielding forage grass [1]. SP has a high protein content, is rich in various essential amino acids and minerals, and contains biologically active substances such as ascorbic acid, phenolic acids, terpenoids, and flavonoids, which may result in resistance to pH reduction in silage [2]. SP is not only suitable for ruminants such as cows and sheep, but also for monogastric animals such as pigs, chickens and ducks, and can even be used as feed for herbivorous fish. Its wide applicability makes SP important in the global livestock industry. SP is highly resilient to a wide range of soil and climatic conditions, including cold, heat, salinity and flooding tolerance. This resilience makes it easier and more feasible to grow and promote SP globally, providing reliable feed security for the livestock industry.

The growth characteristics of SP show a clear seasonal pattern, with vigorous growth and luxuriant foliage in summer, followed by a gradual slowing down of growth in fall and winter, with foliage yellowing and ceasing to grow and an inability to satisfy continuous supply needs throughout the year [3]. Although SP is cold hardy, its growth can still be limited under extreme low temperature conditions. Ensiling is an efficient forage treatment technology for retaining the nutrients and extending the shelf life of forage. However, fresh SP has a higher moisture content and buffering capacity and a lower water-soluble carbohydrate content, resulting in a poor direct silage effect. Xie et al. [4] showed that alfalfa silage alone had very high pH and proportions of ammonia nitrogen to total nitrogen, low lactic acid concentration, and poor fermentation.

During silage preparation, when no exogenous additives are introduced, the microbial community naturally attached to the raw material plays a dominant role in the fermentation process, which often leads to a loss of dry matter, as well as hydrolysis of the proteins in the silage, thus reducing the nutritional value and utilization efficiency of the feed [5]. Lactic acid bacteria preparations are commonly used as promoters of silage fermentation to effectively control and optimize the silage fermentation process and to reduce unnecessary nutrient loss. Lactic acid bacteria can be divided into two types of fermentation according to their metabolites, namely, homofermentative lactic acid bacteria and heterofermentative lactic acid bacteria. *Lactobacillus plantarum* belongs to homofermentative lactic acid bacteria, which can produce a large amount of lactic acid during silage fermentation, rapidly lowering the pH and improving the fermentation quality [6]. It has been shown that *Leymus chinensis* silage has a higher lactic acid content and a lower pH and butyric acid content after adding *L*. *plantarum* compared to the control [7]. *L. buchneri* belongs to heterofermentative lactic acid bacteria, which can produce short-chain fatty acids that inhibit the growth and reproduction of harmful microorganisms such as yeasts and molds, thus improving the aerobic stability of silage. Xu et al. [8] showed that *L. buchneri* improves the aerobic stability of corn silage and reduces the risk of oxidative deterioration of nutrients during the aerobic exposure of silage. However, once silage is exposed to air, the lactic acid accumulated within it is readily utilized by yeasts and other harmful microorganisms in the environment, which, in turn, triggers the aerobic spoilage of silage, leading to reduced feed quality and nutrient loss [9]. To address this issue, relying only on homofermentative lactic acid bacteria may increase the risk of aerobic spoilage. A combination of homofermentative and heterofermentative lactic acid bacteria can improve this phenomenon, and the acetic acid produced by heterofermentative lactic acid bacteria can effectively inhibit the growth of harmful microorganisms [10]. Zhang et al. [11] investigated the effect of *L. plantarum* and *L. buchneri* on alfalfa silage and found that the composite addition group had the highest acetic acid and soluble sugar contents and the lowest ammoniacal nitrogen and yeast contents compared to the group where they were added individually. Meanwhile, under aerobic conditions, the combined addition group had the lowest carbon dioxide and mold contents and the best aerobic stability.

At present, there are no reports on the effects of adding *L. plantarum* and *L. buchneri* alone or in combination on SP-silage. There may be differences in the silage quality and aerobic stability of SP-silage inoculated with different types of fermenting lactic acid bacteria. Therefore, in this experiment, *L. plantarum* and *L. buchneri* were added individually or in combination to SP-silage to investigate the effects of different fermentation types of lactic acid bacteria on the fermentation quality, in vitro digestibility, and aerobic stability of SP-silage, which provided a scientific basis for the subsequent optimization of Lactobacillus combinations in order to improve the quality of silage.

## 2. Materials and Methods

### 2.1. Experimental Materials and Design

SP was provided by Hulin Fengshan Herbal Medicine Planting Farmers’ Specialized Cooperative, Hulin City, and was sown on 16 May 2023, in a high-yield forage base in Tongyu County, Baicheng City, Jilin Province, China (122°2′–123°30′ E longitude, 44°13′–45°16′ N latitude, 170 m above sea level). It was mowed on 20 September, leaving a stubble height of 2–3 cm, and manually mowed and placed in a cool place to wilt overnight so that its moisture modulation was between 65% and 70%. Subsequently, the guillotine was utilized to cut the SP to 1–2 cm, which was then mixed well for silage preparation. LP and LB were purchased from the China Industrial Microbial Strain Preservation and Management Center, and the strain preservation numbers were CICC 6026 and CICC 20293, respectively. The chemical composition, buffering capacity, energy, and microbial population of the silage ingredients are shown in Table 1.

The following additive groups were set up in this experiment: (1) no additive (Control); (2) *Lactobacillus plantarum* (LP, with a viable count of 5 × 10^5^ cfu g^−1^ FW); (3) *Lactobacillus buchneri* (LB, with a viable count of 5 × 10^5^ cfu g^−1^ FW); (4) mixed preparations of *L. plantarum* and *L. buchneri* (LPLB, with a viable count of 5 × 10^5^ cfu g^−1^ FW). On 21 September 2023, SP-silage preparation was carried out at a high-yield forage base in Tongyu County, Baicheng City, Jilin Province, China (122°2′–123°30′ E longitude, 44°13′–45°16′ N latitude, 170 m above sea level). Lactic acid bacteria were all added at 50 mg kg^−1^ FW. The bacterial solution was loaded into a pre-sterilized hand sprayer (DLX-SD-BJ, Ruichen Plant Protection Co., Ltd., Linyi, China) and evenly sprayed on the short-cut raw materials, while the control group was sprayed with an equal amount of distilled water and stirred thoroughly in a plastic container sterilized with 75% ethanol. The silage materials were then packed into 240 mm × 350 mm polyethylene vacuum-packed bags (Zhongzheng Fangxin Packaging Material Co., Ltd., Chengdu, China) of 500 g each, with six replicates per additive group. The first three replicates were used for fermentation quality, chemical composition, energy, and in vitro digestibility analysis, and the remaining replicates were used for aerobic stability testing. Vacuum sealing was then carried out using a vacuum packaging machine (CS-SP, Changsheng Machinery Co., Ltd., Quanzhou, China). SP-silage was prepared by placing the materials at room temperature, protected from light, for 60 days.

### 2.2. Fermentation Quality Analysis and Microbial Counting

After the silage was opened, 20 g of the mixed sample was taken, and 180 mL of distilled water was added before sealing and soaking in a refrigerator at 4 °C for 24 h. The extract was obtained by filtration using 4 layers of gauze and used to determine the subsequent fermentation parameters. The pH of the extract was determined using a pH meter (PHBJ-261L, Qinkun Pump Co., Ltd., Shanghai, China). Ammoniacal nitrogen (NH_3_-N) was determined using the phenol–sodium hypochlorite colorimetric method [12]. The extracts were subjected to 10 min of centrifugation using a bench-top high-speed refrigerated centrifuge (GL21M, Kaida Scientific Instruments Co., Ltd., Changsha, China) and then filtered using 0.22 um aqueous filter membranes, followed by determination of the lactic acid (LA), acetic acid (AA), propionic acid (PA), and butyric acid (BA) contents using a high-performance liquid chromatograph [13].

Another portion of the filtrate was diluted to 10^−3^, 10^−4^, and 10^−5^ with sterile water using the 10-fold dilution method, while the number of lactic acid bacteria, yeasts, and molds in the silage was determined using the plate counting method. Lactic acid bacteria were counted after anaerobic incubation at 37 °C for 48 h using De Man, Rogosa, and Sharpe medium. Yeasts and molds were counted using potato dextrose agar medium at 30 °C for 48 h. The data obtained are expressed as log_10_ cfu g^−1^ FW.

### 2.3. Chemical Composition and Energy Analysis

The silage samples were weighed fresh, placed in a constant temperature blast drying oven at 105 °C for 30 min, dried at 65 °C until constant weight, and weighed dry. The dried samples were pulverized through different mesh sieves and mixed well in self-sealing bags to determine the chemical composition. The dry matter (DM) content was determined using the drying method; the crude ash content was determined using the scorching method at 550 °C; the crude protein (CP) content was determined using a fully automated Dumas Nitrogen Determination Instrument; the crude fat content was determined using the Soxhlet extraction method [14]. The acid detergent fiber (ADF), neutral detergent fiber (NDF), acid detergent lignin (ADL), and crude fiber (CF) contents were determined using an ANKOM Model A2000i Fully Automatic Fiber Meter (Ankom Technology, Fairport, NY, USA) [15]. The water-soluble carbohydrate (WSC) content was determined using the anthrone–sulfuric acid colorimetric method [16]. The buffering capacity (BC) was determined via titration [17]. The gross energy (GE) was determined using a fully automated oxygen bomb calorimeter [14], and then the digestible energy (DE), metabolizable energy (ME), net energy for maintenance (NEm), net energy for weight gain (NEf), and net energy for a lactating cow (NEl) were calculated [18].

### 2.4. Aerobic Stability Analysis

The silage was mixed well and put into a 1 L polyethylene tank and kept at room temperature. The upper layer was sealed with a black plastic sheet and tied with small holes to prevent moisture loss and external contamination and allow air to enter the fermenter freely. At the same time, thermocouples from a multiplexed automatic temperature recorder (RX6000F, Mei-Control Automation Technology Co., Ltd., Hangzhou, China) were inserted in the center of the silage and exposed aerobically for 7 days. The silage and ambient temperatures were recorded automatically at 1 h intervals. The time required for a 2 °C difference between the silage and ambient temperatures was used to assess the aerobic stability of the silage [19].

### 2.5. In Vitro Fermentation Parameter Analysis

This experiment strictly adhered to the Guidelines for Ethical Review of Laboratory Animal Welfare in China and was approved by the Laboratory Animal Welfare Ethics Committee of Jilin University (license no. SY20209600). The rumen fluid was obtained from four healthy small-tailed chilly sheep (mean weight of 35 kg) of similar body conditions fitted with permanent fistulas located at the Agricultural Experimental Base of Jilin University. Sufficient rumen fluid was collected before morning feeding, placed in plastic bottles preheated at 39 °C and filled with carbon dioxide, mixed well, and then filtered through 4 layers of gauze, with the entire operation being strictly anaerobic. Buffers were prepared using the method of McDougall (1948). At the same time, the rumen fluid was mixed with the buffer at a ratio of 1:4 to prepare a mixed culture solution, and carbon dioxide was passed to ensure anaerobic conditions before placing it in a 39 °C water bath for backup. A total of 52 fiber bags were prepared (12 silage samples × 4 parallel samples + 4 blank controls). A 0.5 g silage sample was weighed in a filter bag (ANKOM F57, aperture 25 um, ANKOM Technology Corporation, Macedon, NY, USA) and sealed in a serum bottle (100 mL). Then, 50 mL of the mixed culture solution was added to the serum bottle and passed through carbon dioxide to maintain an anaerobic environment. This was then immediately put on the rubber stopper and aluminum cap, using special sealing pliers to press tightly, and then it was placed into the constant-temperature culture oscillating shaker (39 °C, 100 r min^−1^) to incubate for 48 h. All samples were incubated at once. At the end of the timer, the filter bag in the serum bottle was removed and washed with cold distilled water. After washing, it was placed in an electrically heated constant-temperature blast drying oven (65 °C, 48 h) to dry until constant weight. The filter bag containing the samples was removed from the electrically heated constant-temperature blast-drying oven, and the in vitro dry matter digestibility (IVDMD), in vitro organic matter digestibility (IVOMD), in vitro crude protein digestibility (IVCPD), and in vitro neutral detergent fiber digestibility (IVNDFD) were calculated from the weight loss of the samples before and after incubation [20].

### 2.6. Statistical Analysis

The data were analyzed according to a 2 × 2 factorial treatment design model using the GLM program of SPSS (version 26, International Business Machines Corporation, Armonk, NY, USA):Y_ijk_ = μ + L_j_ + B_k_ + (L × B)_jk_ + ε_ijk_
where Y_ijk_ denotes the response variable, μ is the total mean, L_j_ is a fixed effect of LP 6026 j (j = 1, 2), B_k_ is a fixed effect of LB 20293 k (k = 1, 2), (L × B)_jk_ is a fixed interaction effect of LP 6026 j and LB 20293 k, and ε_ijk_ is the error term.

When at least one interaction was significant (*p* < 0.05), significant differences between means were determined by Tukey’s method, where *p* < 0.05 was recognized as significant.

## 3. Results

### 3.1. Fermentation Quality of Silphium perfoliatum L. Silage

The fermentation quality of SP-silage is shown in Table 2. The pH was significantly lower in all treatment groups compared to the control, with the LP-added SP-silage having the lowest pH of all treatment groups (*p* = 0.002). The LA content of the SP-silage from the three lactic acid bacteria treatment groups was significantly higher than that in the control group (*p* < 0.001). The AA content was highest in the LB group, which was significantly higher than that in the control and LP groups (*p* < 0.001). The BA content of the SP-silage in the LP group was lower than in the other treatment groups, but the difference was not significant (*p* = 0.403). The addition of LP, LB, and LPLB significantly reduced the NH_3_-N/TN in the SP-silage compared to the control (*p* = 0.003).

LP significantly affected the pH, LA, and NH_3_-N/TN (*p* < 0.01). LB had a highly significant effect on LA and AA (*p* < 0.01) and a significant effect on NH_3_-N/TN (*p* < 0.05). There was also an LP × LB interaction effect on the pH, LA, and NH_3_-N/TN (*p* < 0.05). The pH decreased significantly with the addition of LP, with LB having a smaller effect on the SP-silage (4.22 vs. 4.44; *p* = 0.002). The addition of LB elevated the LA content, but LP was better for the SP-silage (4.92 vs. 4.17; *p* < 0.001). Adding LP and LB individually reduced NH_3_-N/TN, but a mixture of LP and LB provided better silage results (2.96 vs. 1.47; *p* = 0.003).

### 3.2. Chemical Composition of Silphium perfoliatum L. Silage

The chemical compositions of SP-silage are shown in Table 3. The addition of LP, LB, and LPLB significantly increased the DM and CP content of the SP-silage compared to the control group, with the LPLB group having the highest DM and CP content (*p* < 0.001). The OM content of the SP-silage from LB and LPLB was significantly higher than that in the control group (*p* = 0.038), but there was no significant difference between the control and LP groups. The NDF and ADF contents were significantly lower in the LB and LPLB groups compared to the control group, with the SP-silage supplemented with LB having the lowest NDF and ADF contents among all treatment groups (*p* = 0.045; *p* < 0.001). The ADL content of the SP-silage in the LB and LPLB groups was significantly lower than that in the control group (*p* = 0.023), but there was no significant difference between the two groups.

LP had a highly significant effect on the DM and CP content (*p* < 0.01). LB had a highly significant effect on the DM and CP, ADF, and ADL contents (*p* < 0.01), as well as a significant effect on the OM and NDF contents (*p* < 0.05). The was also an LP × LB interaction effect on the ADF content (*p* < 0.01). The addition of LP had no significant effect on the ADF content, while the addition of LB significantly reduced it (24.22 vs. 22.61; *p* < 0.001).

### 3.3. Energy of Silphium perfoliatum L. Silage

The energy of the SP-silage is shown in Table 4. The GE and DE of the SP-silage were significantly higher in the LB and LPLB groups than in the control and LP groups (*p* = 0.005; *p* < 0.001), but there was no significant difference between the two groups. The ME was highest in the LB group and significantly higher than in the control and LP groups (*p* = 0.017). The addition of LB and LPLB significantly increased the NEm and NEl of the SP-silage compared to the control (*p* = 0.029; *p* = 0.030). The NEf of the SP-silage was higher in the LB group than in the other treatment groups, but the difference was not significant (*p* = 0.063). LB had a highly significant effect on the GE, DE, ME, NEm, and NEl (*p* < 0.01).

### 3.4. In Vitro Digestibility of Silphium perfoliatum L. Silage

The in vitro digestibility of SP-silage is shown in Table 5. The *IV*DMD and *IV*CPD were significantly higher in all treatment groups compared to the control, with the SP-silage supplemented with LPLB having the highest *IV*DMD and *IV*CPD among all treatment groups (*p* = 0.001; *p* < 0.001). The *IV*NDFD was highest in the LB group, which was significantly higher than in the control and LP groups (*p* = 0.004). The *IV*OMD of the SP-silage in the LPLB group was higher than in the other treatment groups, but the difference was not significant (*p* = 0.164).

LP had a highly significant effect on the *IV*DMD and *IV*CPD (*p* < 0.01). LB had a highly significant effect on the *IV*DMD, *IV*CPD, and *IV*NDFD (*p* < 0.01). There was also an LP × LB interaction effect on the *IV*NDFD (*p* < 0.05). Adding LB provided better silage results than adding LP. However, the best silage results were obtained with a mixture of LP and LB (52.65 vs. 56.14; *p* = 0.004).

### 3.5. Aerobic Stability of Silphium perfoliatum L. Silage

The temperature changes of the SP-silage exposed to aerobic conditions as a result of the additives are shown in Figure 1. The temperature of the three lactic acid bacteria additive groups increased slowly from 0 to 48 h and rapidly from 48 to 96 h, with the increase in temperature being more pronounced in the control and LP groups. The LB group temperature took the longest time to reach 2 °C above ambient, while the LP group temperature took the shortest time.

The time required for the SP-silage temperature to exceed room temperature by 2 °C after exposure to air is shown in Figure 2: 87 h in the control group, 66 h in the LP group, 138 h in the LB group, and 117 h in the LPLB group. The aerobic stability of the SP-silage was significantly reduced by 24.14% in the LP group and increased by 58.62% and 34.48% in the LB and LPLB groups, respectively, compared to the control group (*p* < 0.05).

## 4. Discussion

### 4.1. Effect of Different Fermenting Lactic Acid Bacteria on the Fermentation Quality of Silphium perfoliatum L. Silage

The selection of suitable microbial silage additives can promote the proliferation of lactic acid bacteria, produce LA to make the pH drop rapidly to form an acidic environment suitable for silage fermentation, reduce the loss of nutrients, and improve aerobic stability [21]. In this experiment, different fermentation types of lactic acid bacteria were added to SP-silage to rapidly acidify the fermentation system so as to shorten the fermentation time and improve the fermentation quality [22]. Studies have shown that different fermentation types of lactic acid bacteria positively affect the fermentation quality of silage during the silage process through their degradation of feed substrates and metabolite conversion mechanisms [23].

Silage fermentation is a complex dynamic process involving the mutual transformation and accumulation of many substances. During this process, the pH and LA content within the silage system is a key parameter in measuring the fermentation quality. Generally, the lower the pH and the higher the LA content of silage, the better the fermentation quality [24]. During fermentation, lactic acid bacteria convert sugars to pyruvic acid through glycolytic pathways (e.g., the EMP pathway), which in turn is reduced to LA by lactate dehydrogenase [25]. As the fermentation of lactic acid bacteria continues, LA gradually accumulates in the silage, leading to a significant decrease in the pH of the silage. An acidic environment with low pH has a damaging effect on the cell membranes and enzyme systems of harmful microorganisms, thus inhibiting their growth and reproduction. Also, in addition to the production of LA, lactic acid bacteria may produce other antimicrobial substances (e.g., bacteriocins, hydrogen peroxide, etc.), which have a direct killing or inhibitory effect on harmful microorganisms [25,26]. In this experiment, the addition of LP, LB, and LPLB significantly reduced the pH and significantly increased the LA content of SP-silage compared to the control. This is due to the high activity of exogenously added lactic acid bacteria and the ideal anaerobic environment provided by the compacted silage, which allowed the lactic acid bacteria to proliferate and produce LA. The production of LA further reduced the pH of the silage, creating an acidic environment that inhibited the colonization of other undesirable microorganisms, which, in turn, effectively enhanced the fermentation quality of the silage [27]. Yang et al. [28] showed that the addition of LP significantly increases the LA content, decreases the pH, and improves the fermentation quality in alfalfa silage, which is consistent with the results of this experiment. Romero et al. [29] showed that the addition of LB to silage increases the AA content and LA concentration than the control without any lactic acid bacteria addition. In addition, the LPLB group had the highest LA content in this experiment, and both the LB and LPLB groups had a higher AA content, which is consistent with the findings of Weinters et al.’s study [30]. This was because the homofermentative lactic acid bacteria fermentation end product was LA, whereas the heterofermentative lactic acid bacteria were able to break LA down into AA, 1,2-propanediol, carbon dioxide, and ethanol in an anaerobic environment. BA is a product formed by harmful bacteria during silage fermentation by decomposing and transforming AA and sugar, and a high content of BA represents a strong activity of harmful bacteria, which is unfavorable to silage fermentation. In this experiment, the BA content of all three additive groups was lower than that in the control group, which indicates that the addition of different fermentation types of lactic acid bacteria was effective in improving the fermentation quality of SP-silage.

Spoilage microorganisms decompose proteins and amino acids in silage to produce NH_3_-N. In this experiment, all three additive groups significantly reduced the NH_3_-N content of SP-silage, indicating that the addition of the three lactic acid bacteria significantly inhibited the protein decomposition of the silage. In SP-silage inoculated with three species of lactic acid bacteria, the pH of the silage was reduced, leading to inhibition of plant protease activity, which, in turn, reduced protein and amino acid catabolism and decreased NH_3_-N production [31]. Blajman et al. [32] added a lactic acid bacteria inoculant to alfalfa silage, which reduced the pH and NH_3_-N content of the silage compared to the control, in agreement with the results of the present experiment. Filya et al. [33] showed that the combination of LP and LB is effective in lowering the pH of silage and reducing the concentration of NH_3_-N, thus reducing nutrient loss during fermentation. The pH of the LPLB group was lower than that in the LB group but higher than that in the LP group, with the highest LA content and the lowest NH_3_-N/TN. This may be due to the presence of synergistic effects, as evidenced by the LP × LB interaction effect on pH, LA, and NH_3_-N/TN (Table 2). LP produced LA, which reduced the pH and inhibited the growth of harmful bacteria. LB promoted the conversion of LA to AA, inhibited the activity of aerobic microorganisms, and slowed down the decomposition of amino acids, indicating that the combination of different fermentation types of lactic acid bacteria was more efficient in inhibiting the activity of harmful microorganisms.

### 4.2. Effect of Different Fermenting Lactic Acid Bacteria on the Energy and Nutritional Value of Silphium perfoliatum L. Silage

Energy is one of the most important indicators for assessing the nutritional value of feeds and is the basis for the study of ME and DE in animals. In this experiment, the LB and LPLB groups significantly increased the GE, DE, ME, and NEl of SP-silage, indicating that the addition of LB and LPLB is an effective method to promote the energy accumulation of silage. This may be because the addition of LB and LPLB inhibits the aerobic spoilage bacterial activity and reduces the consumption of nutrients in SP-silage.

The nutritional content is one of the most important indicators for determining the quality of SP-silage. After silage, the nutrients in SP were reduced compared to pre-silage, which is mainly due to the involvement of a variety of microorganisms in the silage fermentation process. Under the joint action of these microorganisms, nutrients in the substrate were consumed in large quantities, such as CP, which was converted to NH_3_-N and amines. In this experiment, the three lactic acid bacteria groups significantly increased the levels of DM and CP compared to the control group, with the LPLB group having the highest levels. This result indicates that both homofermentative and heterofermentative lactic acid bacteria are effective in inhibiting the degradation of DM and CP in SP-silage. In addition, the addition of a mixture of different fermentation types of lactic acid bacteria showed better silage results compared to single strains, which is in line with the findings of Contreras-Govea et al. [34]. This is because the addition of lactic acid bacteria accelerated the fermentation process, leading to a rapid drop in pH, which effectively inhibited the activity of aerobic microorganisms and proteolytic enzymes, thus reducing nutrient loss [35]. In addition, the LB and LPLB groups were more effective than the LP group in enhancing the DM and CP contents, which may be related to the difference in the fermentation end-products of LB as a heterofermentative lactic acid bacteria. The reason may be that LB produces ethanol and AA during the decomposition of LA. Ethanol has a fungicidal effect, while AA inhibits the reproduction of some fungi. Compared to LP, LB was much better at retaining DM and CP. Therefore, the addition of LB and LPLB is effective in reducing the loss of DM and CP in SP-silage, which is in agreement with the findings of Reich et al. [36].

NDF and ADF are key nutritional parameters for forage quality. A low NDF content indicates good feed palatability, while a low ADF content implies high feed utilization. Studies have shown that the addition of different fermentation types of lactic acid bacteria to barley, spent mushroom substrate, and perennial ryegrass results in a decrease in the levels of NDF and ADF, which is in agreement with the results of the present experiment [37,38,39]. The was also an LP × LB interaction effect on the ADF content (*p* = 0.003). In SP-silage inoculated with LP alone, there was no significant change in the content of ADF compared to the control. LB and LPLB were particularly effective in reducing the NDF and ADF contents. This may be due to the production of ferulic acid esterase during LB reproduction. Ferulic acid esterase breaks down the ferulic ester bond between lignin and hemicellulose, destroying the fiber structure, which, in turn, reduces the fiber content and enhances the fiber digestibility of the feed to some extent [39].

### 4.3. Effect of Different Fermenting Lactic Acid Bacteria on the In Vitro Digestibility of Silphium perfoliatum L. Silage

In vitro simulation of rumen fermentation provides a more accurate estimate of the potential digestibility of forage in ruminants [40]. By determining the amount of *IV*DMD in the rumen, it is possible to estimate the energy distribution potential in the feed [41]. The results of this experiment showed that the addition of different fermentation types of lactic acid bacteria to SP-silage significantly increased the *IV*DMD and *IV*CPD. Tian et al. [42] found that the introduction of lactic acid bacteria in *Leymus chinensis* silage results in significantly higher *IV*DMD and *IV*CPD compared to the control. This finding is supported by Wan et al. [43], where the addition of lactic acid bacteria to wilted sudangrass silage similarly elevated the *IV*DMD and *IV*CPD, consistent with the results of this experiment. This is due to the production of LA by lactic acid bacteria to form an acidic environment, which inhibits the activity of harmful microorganisms and reduces nutrient consumption, resulting in an increase in the *IV*DMD and *IV*CPD. Research has shown that feeding animals silage supplemented with lactic acid bacteria improves animal performance to some extent. Nkosi et al. [44] showed that lambs fed whole-crop maize silage supplemented with *L. buchneri NCIMB 40788* improve their crude protein intake, final body weight, and average daily gain compared to the control. Inoculation of alfalfa silage with *L. buchneri 40788* results in increased milk production in lactating dairy cows compared to the control [45]. Addah et al. [46] found that barley silage fed to growing cattle with the addition of a composite fungicide (*L. buchneri LN4017* + *L. plantarum LP7109* + *L. casei LC3200*) is superior to the control in terms of feed conversion ratio and average daily weight gain, suggesting that the compound bacterial agent has a positive effect on improving the performance of animal production. Lactic acid bacteria in the fermentation process can degrade the large molecule proteins in silage into free amino acids and small-molecule polypeptides, providing nutrients for the organism and promoting digestion and absorption in the gastrointestinal tract. Lactic acid bacteria can also secrete and produce a variety of digestive enzymes in the host body to decompose and transform antinutritional factors, thus improving the production performance of animals [47]. LP and LB, as probiotics, can colonize and multiply in the animal’s intestinal tract to form a dominant flora. They inhibit the growth of harmful microorganisms such as E. coli and Salmonella, promote intestinal health, reduce the incidence of gastrointestinal diseases, and increase economic benefits by reducing deaths and treatment costs due to disease. At the same time, the combined addition of LP and LB to SP-silage improved the nutrient utilization of silage and reduced the animal’s feed requirement, thus reducing feeding costs. Therefore, based on the results of this experimental study, we can expect that if SP-silage with added LPLB was introduced into ruminant diets, it could effectively make up for the deficiencies of a single inoculant and contribute to the improvement of ruminant production performance and economic benefits for farmers. In addition, the combined addition of LP and LB to SP-silage needs to take into account the fact that lactic acid bacteria may be affected by temperature, humidity, pH and other factors during the silage process, leading to a decrease in survival. Faced with such a problem, we can optimize the silage conditions, such as controlling the appropriate temperature and humidity, and adding the right amount of sugar substances as a fermentation substrate to improve the survival rate of lactic acid bacteria.

The fibrous matter content and composition of feed are important factors affecting the internal environment of the rumen in ruminants. Herein, the NDF content of feeds affected the *IV*NDFD, which led to different *IV*NDFDs for the different feeds. In addition, there was an LP × LB interaction effect on the *IV*NDFD, suggesting that the combined synergistic effect of LP and LB significantly improved the in vitro digestibility of the silage. Although the addition of either LP or LB alone increased the *IV*NDFD of the silage, the LPLB group had the highest *IV*NDFD. Weinberg et al. [48] found that the addition of specific microbial formulations to silage may lead to a decrease in LA levels due to competition with rumen microorganisms, further contributing to an increase in rumen pH and providing a more favorable environment for the growth of fibrolytic bacteria in the rumen. Therefore, inoculation of silage with microbial preparations can improve fiber digestion in the rumen. Studies have shown that the addition of LB to corn silage improves the *IV*NDFD, consistent with the results of this experiment [48]. Nsereko et al. [39] added lactic acid bacteria with ferulic acid esterase expression to silage and found them to be effective in improving the fiber digestibility of silage. Ferulic acid esterase is an enzyme capable of hydrolyzing ferulic acid ester bonds, which are widely found in non-starch polysaccharides (e.g., hemicellulose) in plant cell walls. The possible mechanism of action of this enzyme lies in the weakening of the structural stability of the cell wall by acting on the ferulic acid ester bond in hemicellulose [39]. At the same time, ferulic acid esterase strips hemicellulose from lignin, thereby increasing the fiber digestibility of the feed. Kang et al. [49] showed that *L. buchneri PTA6138* has ferulic acid esterase activity. When *L. buchneri PTA6138* was used in combination with *L. casei PTA6315*, it was effective in improving the *IV*NDFD of corn silage. In this assay, we did not verify whether LB had ferulic acid esterase activity. However, both LB and LPLB were observed to increase the *IV*NDFD of SP-silage. Based on this finding, we believe that a more in-depth study of this phenomenon is warranted.

### 4.4. Effect of Different Fermenting Lactic Acid Bacteria on the Aerobic Stability of Silphium perfoliatum L. Silage 

Once the silage bag was opened, the original anaerobic state was rapidly transformed into an aerobic state, which, in turn, activated the activity of aerobic microorganisms. In general, spoilage of silage is mainly due to the activity of aerobic microorganisms such as yeasts and molds in an aerobic environment. Yeasts and molds utilize the LA, remaining sugars, and CP in the silage after it is opened, triggering a continuous release of heat from the silage, which ultimately leads to spoilage of the silage, a phenomenon confirmed in the study of Hu et al. [50].

In this experiment, the aerobic stability of the LP group was significantly lower than that in the control group, while the aerobic stability of the LB and LPLB groups was significantly higher, which was prolonged by 51 and 30 h, respectively, compared to the control group. During fermentation, homofermentative lactic acid bacteria dominated the process, producing LA as the main fermentation product. Short-chain fatty acids inhibited the growth of yeast and mold. Wang et al. [51] showed that LA produced by homofermentative lactic acid bacteria can be used as a substrate for yeast growth under aerobic exposure conditions while producing fewer short-chain fatty acids. In addition, LA itself is not an effective antimould agent. Consequently, SP-silage inoculated with homofermentative lactic acid bacteria is more susceptible to spoilage during silage storage, which has been confirmed in previous studies [52]. Weinberg et al. [35] showed that LP reduces the aerobic stability of wheat silage and is highly susceptible to spoilage, which is in agreement with the results of this experiment. Heterofermentative lactic acid bacteria, after becoming the dominant flora in silage, improve the aerobic stability of silage by increasing the concentration of AA and 1,2-propanediol and producing bacteriocins [53]. AA has a strong fungus-inhibiting ability and can prevent feed spoilage. In this experiment, both the LB and LPLB groups had a higher AA content, which was significantly higher than the control and LP groups. This is one of the main reasons why the LB and LPLB groups were able to improve the aerobic stability of SP-silage [54]. According to Wang et al. [51], inoculation with either LB or LPLB significantly improves the aerobic stability of whipgrass silage. Filya et al. [33] showed that wheat, sorghum, and corn silages supplemented with LB alone have a lower WSC, LA content, and pH compared to LP alone, yet they are more aerobically stable. In addition, they found that the addition of LPLB not only helps to reduce the loss of DM but also enhances the aerobic stability of silage, a finding that coincides with the results of this experiment.

## 5. Conclusions

The addition of LP and LB alone or in combination improved the silage quality and in vitro digestibility of SP-silage, with the best silage results obtained using a LPLB inoculant. The LB inoculant alone can improve the aerobic stability of SP-silage. Further in vivo trials are needed to validate and evaluate the actual effects of LPLB-inoculated SP-silage on ruminant growth performance before recommending our findings to farmers.

## Figures and Tables

**Figure 1 animals-14-02279-f001:**
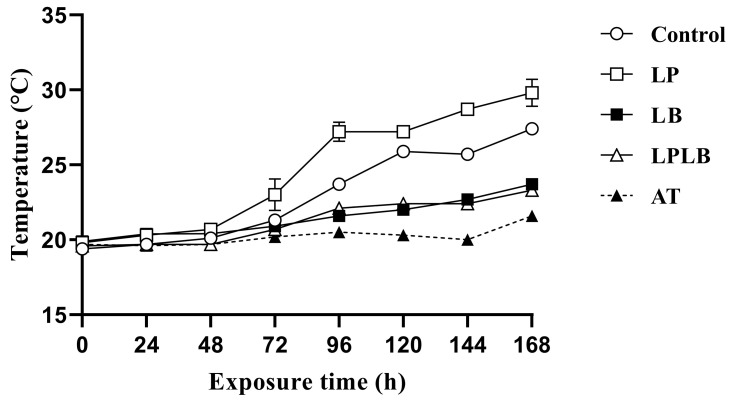
Effect of the application of additives on the temperature of *Silphium perfoliatum* L. silage after exposure to aerobic conditions. Control, no additive; LP, *Lactobacillus plantarum*; LB, *Lactobacillus buchneri*; LPLB, mixture of *Lactobacillus plantarum* and *Lactobacillus buchneri*; and AT, ambient temperature.

**Figure 2 animals-14-02279-f002:**
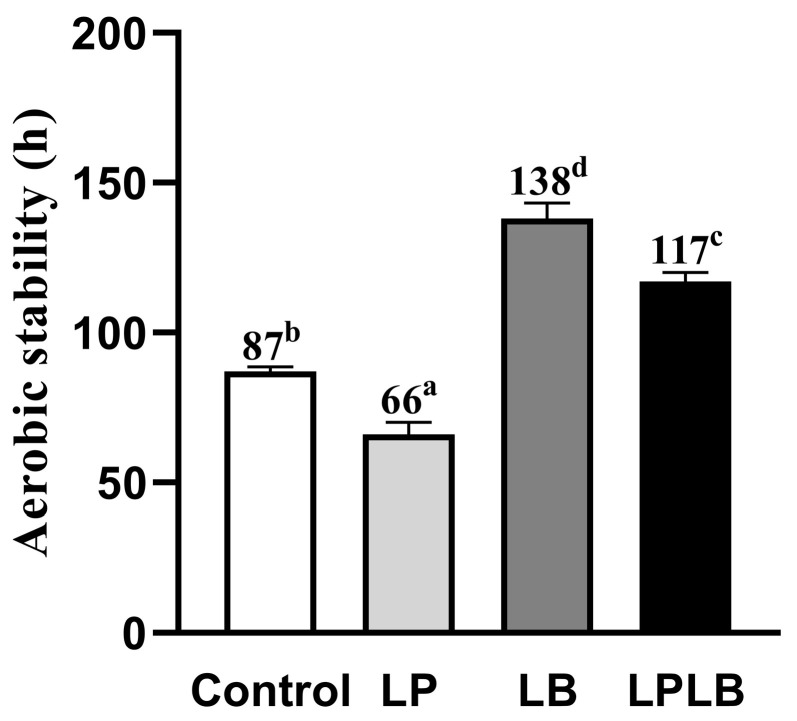
Time required for *Silphium perfoliatum* L. silage temperature to exceed room temperature by 2 °C after exposure to air. Control, no additive; LP, *Lactobacillus plantarum*; LB, *Lactobacillus buchneri*; LPLB, mixture of *Lactobacillus plantarum* and *Lactobacillus buchneri*; ^a–d^ Means of the additive treatments with significant differences between superscript letters (*p* < 0.05).

**Table 1 animals-14-02279-t001:** Chemical composition, buffering capacity, energy, and microbial counts of *Silphium perfoliatum* L.

Items ^‡^	*Silphium perfoliatum* L.
Chemical composition and buffering capacity	
Dry matter (% FW)	31.43
Organic matter (% DM)	82.26
Crude protein (% DM)	22.97
Neutral detergent fiber (% DM)	61.50
Acid detergent fiber (% DM)	26.18
Acid detergent lignin (% DM)	9.43
Water-soluble carbohydrate (% DM)	4.68
Buffering capacity (mEq kg^−1^ DM)	242.21
Energy	
GE (MJ kg^−1^ DM)	15.24
DE (MJ kg^−1^ DM)	11.70
ME (MJ kg^−1^ DM)	8.72
NEm (M Jkg^−1^ DM)	6.24
NEl (MJ kg^−1^ DM)	5.33
NEf (MJ kg^−1^ DM)	4.61
Microbial counts	
Lactic acid bacteria (log_10_ cfu g^−1^ FW)	4.31
Yeasts (log_10_ cfu g^−1^ FW)	3.06
Moulds (log_10_ cfu g^−1^ FW)	1.29

^‡^ FW, fresh weight; DM, dry matter; GE, gross energy; DE, digestible energy; ME, metabolizable energy; NEm, net energy for maintenance; NEl, net energy for lactating cow; NEf, net energy for weight gain; and cfu, colony-forming units.

**Table 2 animals-14-02279-t002:** Fermentation quality of *Silphium perfoliatum* L. silage prepared with lactic acid bacteria.

Items ^‡^	Additives ^†^				SEM	*p*-Value	Significance of Main Effects and Interactions		
	Control	LP	LB	LPLB			LP	LB	LP × LB
pH	4.64 ^c^	4.22 ^a^	4.44 ^b^	4.33 ^a b^	0.071	0.002	0.001	0.399	0.014
LA (%DM)	2.63 ^a^	4.92 ^c^	4.17 ^b^	5.51 ^c^	0.279	<0.001	<0.001	0.001	0.044
AA (%DM)	0.45 ^a^	0.47 ^a^	0.90 ^b^	0.86 ^b^	0.049	<0.001	0.676	<0.001	0.437
PA (%DM)	0.01	0.02	0.03	0.02	0.008	0.064	0.780	0.062	0.078
BA (%DM)	0.49	0.25	0.33	0.32	0.138	0.403	0.235	0.680	0.259
NH_3_-N (%TN)	2.96 ^b^	1.60 ^a^	1.76 ^a^	1.47 ^a^	0.284	0.003	0.003	0.011	0.029

^a–c^ Means of the additive treatments within a row with significant differences between the superscript letters (*p* < 0.05). SEM, standard error of the mean. ^†^ LB, *Lactobacillus plantarum*; LB, *Lactobacillus buchneri*; LPLB, mixture of *Lactobacillus plantarum* and *Lactobacillus buchneri*. ^‡^ DM, dry matter; LA, lactic acid; AA, acetic acid; PA, propionic acid; BA, butyric acid; NH_3_-N, ammonia nitrogen; and TN, total nitrogen.

**Table 3 animals-14-02279-t003:** Chemical compositions of *Silphium perfoliatum* L. silage prepared with lactic acid bacteria.

Items ^‡^	Additives ^†^				SEM	*p*-Value	Significance of Main Effects and Interactions		
	Control	LP	LB	LPLB			LP	LB	LP × LB
DM (%FW)	24.68 ^a^	26.69 ^b^	27.38 ^b^	29.47 ^c^	0.525	<0.001	0.001	<0.001	0.924
OM (%DM)	77.36 ^a^	78.06 ^a b^	78.34 ^b^	78.42 ^b^	0.320	0.038	0.123	0.018	0.207
CP (%DM)	16.76 ^a^	18.21 ^b^	21.03 ^c^	21.79 ^d^	0.310	<0.001	0.001	<0.001	0.162
NDF (%DM)	60.55 ^b^	57.40 ^a b^	54.58 ^a^	55.59 ^a^	1.799	0.045	0.424	0.016	0.141
ADF (%DM)	24.81 ^c^	24.22 ^b c^	22.61 ^a^	23.70 ^b^	0.283	<0.001	0.247	<0.001	0.003
ADL (%DM)	7.39 ^b^	6.67 ^a b^	6.10 ^a^	6.08 ^a^	0.371	0.023	0.200	0.007	0.219

^a–d^ Means of the additive treatments within a row with significant differences between superscript letters (*p* < 0.05). SEM, standard error of the mean. ^†^ LP, *Lactobacillus plantarum*; LB, *Lactobacillus buchneri*; LPLB, mixture of *Lactobacillus plantarum* and *Lactobacillus buchneri*. ^‡^ FW, fresh weight; DM, dry matter; OM, organic matter; CP, crude protein; NDF, neutral detergent fiber; ADF, acid detergent fiber; and ADL, acid detergent lignin.

**Table 4 animals-14-02279-t004:** Energy of *Silphium perfoliatum* L. silage prepared with lactic acid bacteria.

Items ^‡^	Additives ^†^				SEM	*p*-Value	Significance of Main Effects and Interactions		
	Control	LP	LB	LPLB			LP	LB	LP × LB
GE (MJ kg ^−1^DM)	15.74 ^a^	15.77 ^a^	16.24 ^b^	16.22 ^b^	0.068	<0.001	0.919	<0.001	0.570
DE (MJ kg ^−1^ DM)	11.85 ^a^	12.07 ^a^	12.52 ^b^	12.64 ^b^	0.168	0.005	0.184	0.001	0.666
ME (MJ kg ^−1^ DM)	9.47 ^a^	9.61 ^a b^	9.98 ^c^	9.90 ^b c^	0.136	0.017	0.726	0.003	0.294
NEm (MJ kg ^−1^ DM)	6.88 ^a^	7.01 ^a b^	7.29 ^b^	7.22 ^b^	0.118	0.029	0.701	0.006	0.266
NEl (MJ kg ^−1^ DM)	5.75 ^a^	5.86 ^a b^	6.04 ^b^	6.10 ^b^	0.100	0.030	0.272	0.006	0.715
NEf (MJ kg ^−1^ DM)	4.49	4.64	4.85	4.78	0.116	0.063	0.653	0.167	0.237

^a–c^ Means of the additive treatments within a row with significant differences between superscript letters (*p* < 0.05). SEM, standard error of the mean. ^†^ LP, *Lactobacillus plantarum*; LB, *Lactobacillus buchneri*; LPLB, mixture of *Lactobacillus plantarum* and *Lactobacillus buchneri*. ^‡^ DM, dry matter; GE, gross energy; DE, digestible energy; ME, metabolizable energy; NEm, net energy for maintenance; NEl, net energy for lactating cow; and NEf, net energy for weight gain.

**Table 5 animals-14-02279-t005:** In vitro digestibility of *Silphium perfoliatum* L. silage prepared with lactic acid bacteria.

Items ^‡^	Additives ^†^				SEM	*p*-Value	Significance of Main Effects and Interactions		
	Control	LP	LB	LPLB			LP	LB	LP × LB
*IV*DMD (%DM)	63.47 ^a^	64.45 ^b^	64.33 ^b^	65.51 ^c^	0.304	0.001	0.001	0.002	0.643
*IV*OMD (%DM)	67.32	68.19	68.14	68.45	0.465	0.164	0.112	0.136	0.428
*IV*CPD (%DM)	60.13 ^a^	61.69 ^b^	64.19 ^c^	64.98 ^d^	0.240	<0.001	<0.001	<0.001	0.059
*IV*NDFD (%DM)	52.65 ^a^	54.45 ^b^	55.44 ^b c^	56.14 ^c^	0.654	0.004	0.272	0.001	0.027

^a–d^ Means of the additive treatments within a row with significant differences between superscript letters (*p* < 0.05). SEM, standard error of the mean. ^†^ LP, *Lactobacillus plantarum*; LB, *Lactobacillus buchneri*; LPLB, mixture of *Lactobacillus plantarum* and *Lactobacillus buchneri*. ^‡^ IVDMD, in vitro dry matter digestibility; IVOMD, in vitro organic matter digestibility; IVCPD, in vitro crude protein digestibility; and IVNDFD, in vitro neutral detergent fiber digestibility.

## Data Availability

The original contributions presented in the study are included in the article, further inquiries can be directed to the corresponding author.

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
