# Peer review of "The Effects of Lactobacillus plantarum and Lactobacillus buchneri on the Fermentation Quality, In Vitro Digestibility, and Aerobic Stability of Silphium perfoliatum L. Silage"

_animals, 2024, doi:10.3390/ani14152279_

Round 1

Reviewer 1 Report

Comments and Suggestions for Authors

The idea of the work is good, but it presents some weaknesses that need to be addressed. The authors should avoid repeating keywords from the title.

Introduction

The introduction needs improvement. I missed a review of the literature addressing the isolated and combined effects of the two bacterial strains, even if in other grasses. This is important to contextualize and justify the study. Additionally, is Silphium a legume? From what the authors report, it does not seem to be a grass. It is essential to clarify this classification. What are the observations in the literature about the fermentation profile of this plant when ensiled exclusively? The introduction suggests that this plant is rich in proteins and amino acids, which may result in resistance to pH reduction in silage. This needs to be clear in the introduction.

Materials and Methods

  • Line 183: Were the animals fed the silages? If so, this should be explicit, as it may affect the in vitro degradation kinetics.
  • Table 1: This table should be included in the materials and methods section as the composition of the original material. It makes no sense to present these results without statistical analysis and comparisons. Therefore, this table should not be in the results section.

Results and discussion

Where are the results for the fermentation profile, such as microbial populations at 60 days of fermentation? I missed the data on water-soluble carbohydrates, as well as dry matter losses, effluents, and gases. Without these data, the work is restricted to chemical composition. The microbial profile data and all the mentioned data should be included in this article.

  • Line 231: The authors should highlight the main values. As written, the reader is dependent on referring to the tables.
  • Line 352: Providing microbiology data could strengthen the authors' argument.
  • Line 356: This statement is unacceptable. When working with microbial additives, it is expected that the conditions enhance bacterial activity to produce high-quality silage. The authors cannot ignore the raw material data. Every plant has an epiphytic bacterial population, but this is not decisive for the fermentation process.
  • Lines 359-361: Is this the only purpose of working with microorganisms? What is expected from microbial inoculation? What type of fermentation? Based on the acid values, it seems there was no control.

The entire discussion section needs improvement.

Conclusion

The conclusion needs to be synthesized and improved, avoiding presenting values except for recommendations.

Author Response

Dear reviewer, thank you for your encouraging and warm comments and suggestions, all of your suggestions are very important, and they all have important guiding significance for our future research work. Based on this we have revised and (we think) strengthened our paper.

Point 1: Additionally, is Silphium a legume? From what the authors report, it does not seem to be a grass. It is essential to clarify this classification. What are the observations in the literature about the fermentation profile of this plant when ensiled exclusively? The introduction suggests that this plant is rich in proteins and amino acids, which may result in resistance to pH reduction in silage. This needs to be clear in the introduction.

Response 1: We fully appreciate your suggestion. We revised that section in the paper. Silphium perfoliatum L. (SP) is not a legume, but a perennial dicotyledonous herb belonging to the genus loosestrife in the family Asteraceae. There are fewer studies on the fermentation of SP when silaged alone. Alfalfa is also a dicotyledonous herb, and based on Xie et al. (2021), it is possible to speculate on the fermentation of SP when silaged alone (Extremely high pH and proportions of ammonia nitrogen to total nitrogen, low lactic acid concentration, poor fermentation. https://doi.org/10.3390/agronomy11010091). (PDF: 56-58, 62, 76-78)

Point 2: Line 183: Were the animals fed the silages? If so, this should be explicit, as it may affect the in vitro degradation kinetics.

Response 2: We fully appreciate your suggestion. In this study, the animals did not eat the experimentally prepared silage.

Point 3: Table 1: This table should be included in the materials and methods section as the composition of the original material. It makes no sense to present these results without statistical analysis and comparisons. Therefore, this table should not be in the results section.

Response 3: We fully appreciate your suggestion. We revised that section in the paper. (PDF:135 )

Point 4: Where are the results for the fermentation profile, such as microbial populations at 60 days of fermentation? I missed the data on water-soluble carbohydrates, as well as dry matter losses, effluents, and gases. Without these data, the work is restricted to chemical composition. The microbial profile data and all the mentioned data should be included in this article.

Response 4: We fully appreciate your suggestion. Data on microbial population, water-soluble carbohydrates, as well as dry matter losses, effluents and gases at 60 days of fermentation were not measured in this experiment. Thank you again for the reminder that in future studies we will first consider determining data on microbial population, water-soluble carbohydrates, as well as dry matter losses, effluents and gases at 60 days of fermentation

Point 5: Line 231: The authors should highlight the main values. As written, the reader is dependent on referring to the tables.

Response 5: We fully appreciate your suggestion. We revised that section in the paper. (PDF: 246, 272, 297, 313)

Point 6: Line 352: Providing microbiology data could strengthen the authors' argument.

Response 6: We fully appreciate your suggestion. The microbial population at 60 days of fermentation was not determined in this experiment. Thank you again for the reminder that in future studies we will first consider determining the microbial population at 60 days of fermentation in order to better strengthen our argument that the selection of suitable microbial silage additives promotes the proliferation of lactic acid bacteria.

Point 7: Line 356: This statement is unacceptable. When working with microbial additives, it is expected that the conditions enhance bacterial activity to produce high-quality silage. The authors cannot ignore the raw material data. Every plant has an epiphytic bacterial population, but this is not decisive for the fermentation process.

Response 7: We fully appreciate your suggestion. In the paper we deleted that section. (PDF: 362-366)

Point 8: Lines 359-361: Is this the only purpose of working with microorganisms? What is expected from microbial inoculation? What type of fermentation? Based on the acid values, it seems there was no control.

Response 8: We fully appreciate your suggestion. This is not the sole purpose of studying microorganisms; that section only addresses fermentation quality. The expected effect of microbial inoculation is to improve the fermentation quality, nutrient content, in vitro digestibility and aerobic stability of SP silage. Lactobacillus plantarum is a homofermentative lactic acid bacteria and Lactobacillus buchneri is a heterofermentative lactic acid bacteria. In this experiment, the addition of LP, LB, and LPLB significantly increased the LA content of SP-silage compared to the control. This is due to the high activity of exogenously added lactic acid bacteria and the ideal anaerobic environment provided by the compacted silage, which allowed the lactic acid bacteria to proliferate and produce LA.

Point 9: The conclusion needs to be synthesized and improved, avoiding presenting values except for recommendations.

Response 9: We fully appreciate your suggestion. We revised that section in the paper. (PDF: 581-586)

Thank you again for your suggestions and hope to learn more from you.

Reviewer 2 Report

Comments and Suggestions for Authors

Comments on the Quality of English Language

Author Response

Dear reviewer, thank you for your encouraging and warm comments and suggestions, all of your suggestions are very important, and they all have important guiding significance for our future research work. Based on this we have revised and (we think) strengthened our paper.

Point 1: Title: The Effects →The effects

Response 1: We fully appreciate your suggestion. We revised that section in the paper. (PDF:2)

Point 2: Order of Keywords: Keywords should be in alphabetical order?

Response 2: We fully appreciate your suggestion. We revised that section in the paper. (PDF:44-46)

Point 3: Line 96 -97: Meanwhile, under aerobic conditions, the group had the lowest carbon dioxide and mold contents and the best aerobic stability.: the group is unclear, therefore, "the combined addition group" is easy to understand.

Response 3: We fully appreciate your suggestion. We revised that section in the paper. (PDF:108)

Point 4: Line 164: The cater-soluble carbohydrate (WSC) content→The water-soluble carbohydrate(WSC) content.

Response 4: We fully appreciate your suggestion. We revised that section in the paper. (PDF:188)

Point 5: Line 232: The fermentation quality of Silphium perfoliatum L. silage → The fermentation quality of SP silage, since the abbreviation of plant name was used. This comment was the same as inLines 258, 283, and 298.

Response 5: We fully appreciate your suggestion. We revised that section in the paper. (PDF:247, 273, 298, 314)

Point 6: Line 308 - 309: with a mixture of LP and LB (52.65 vs. 56.44; p= 0.004). → (52.65 vs. 56.14; p=0.004).

Response 6: We fully appreciate your suggestion. We revised that section in the paper. (PDF:326)

Point 7: Line 345: It is better to explain "Control, no additive,"

Response 7: We fully appreciate your suggestion. We revised that section in the paper. (PDF:353)

Point 8: Figure A: Line style of AT is better to change the broken line, since the ambient temperature iseasy to distinguish from the temperature of the plots.

Response 8: We fully appreciate your suggestion. We revised that section in the paper. (PDF:345)

Point 9: Lines 357 - 358: which was lower than the minimum standard to ensure good silage fermentation (5 log10 cfu gl FW) [21]. As the characteristics of raw material of SP, please refer to the those ofthe other dicotyledonous plants, such as alfalfa. - Since you often referred to the silage fermentation quality of alfalfa. Then, the readers are easy to understand the properties of SP forsilage processing.

Response 9: We fully appreciate your suggestion. We revised that section in the paper. (PDF:362-366)

Point 10: Line 548: using a mixture of L. plantarum and L. buchneri. L. buchneri can be used as an additive - -L. buchneri alone can be used as an additive --

Response 10: We fully appreciate your suggestion. We revised that section in the paper. (PDF:583)

Thank you again for your suggestions and hope to learn more from you.

Reviewer 3 Report

Comments and Suggestions for Authors

This article investigated the effect of two lactic acid bacteria (L. plantarum and L. buchneri) on fermentation quality, in vitro digestibility and aerobic stability of silage from silverleaf grass (Silphium perfoliatum L.). The manuscript is an original and well-done research on silage production. The results of the study fill the gaps in the existing literature, have strong originality and potential practical application value, and provide scientific basis and technical support for obtaining high quality silage from Silphium perfoliatum.However, I think there are still a few shortcomings in the current manuscript.

1.         The introductory section provides background information on silverleaf grass (Silphium perfoliatum L.) as a forage resource, and also mentions the importance of SP for animal husbandry and the severity of the current feed supply problem in China. The authors could also further expand on the potential and importance of silverleaf grass as a feed in a global context.

2.         The introductory section could be more specific in describing the advantages and current challenges (e.g., seasonal growth, poor direct silage results, etc.) of Silphium perfoliatum L. (SP) as a potential forage resource to further emphasize the urgency of the research.

3.         There are many species of lactic acid bacteria that can influence the fermentation quality of silage, why were Lactobacillus plantarum and Lactobacillus buchneri selected for this study and by what mechanism might they act during the silage process?

4.         The introduction section needs to highlight more clearly the innovation and contribution of this study to the existing body of knowledge, as well as its potential impact on the livestock and feed processing industrie.

5.         Authors need to add time and location of silage fermentation and location of fistula sheep to the material approach.

6.         The results section Figure 3.5A is stretched out of shape and the authors need to correct it by adjusting the size and style. Also, the authors need to label the results for significance of differences.

7.         For observed experimental phenomena, the discussion section should go deeper into the biological mechanisms behind them. The authors may cite relevant literature to explain how lactic acid bacteria inhibit the growth of harmful microorganisms by lowering pH through the production of lactic acid, or discuss how lactic acid bacteria affect the degradation of plant cell walls, which in turn affects the in vitro digestibility of feed.

8.         The discussion section should give more consideration to the practical application value of the research results. The potential impact of the combined addition of Lactobacillus plantarum and Lactobacillus buchneri to Silphium perfoliatum L. silage on the productivity and economic benefits of animal husbandry could be explored. Also, problems and solutions that may be encountered in practical applications can be proposed.

Author Response

Dear reviewer, thank you for your encouraging and warm comments and suggestions, all of your suggestions are very important, and they all have important guiding significance for our future research work. Based on this we have revised and (we think) strengthened our paper.

Point 1: The introductory section provides background information on silverleaf grass (Silphium perfoliatum L.) as a forage resource, and also mentions the importance of SP for animal husbandry and the severity of the current feed supply problem in China. The authors could also further expand on the potential and importance of silverleaf grass as a feed in a global context.

Response 1: We fully appreciate your suggestion. We revised that section in the paper. (PDF:62-68)

Point 2: The introductory section could be more specific in describing the advantages and current challenges (e.g., seasonal growth, poor direct silage results, etc.) of Silphium perfoliatum L. (SP) as a potential forage resource to further emphasize the urgency of the research.

Response 2: We fully appreciate your suggestion. We revised that section in the paper. (PDF:62-68, 72-73)

Point 3: There are many species of lactic acid bacteria that can influence the fermentation quality of silage, why were Lactobacillus plantarum and Lactobacillus buchneri selected for this study and by what mechanism might they act during the silage process?

Response 3: The aim of this study was to investigate the effects of different fermentation types of lactic acid bacteria on the fermentation quality, in vitro digestibility and aerobic stability of SP silage, with a view to providing certain scientific basis and technical support for obtaining high quality SP silage in production. Lactobacillus plantarum is a homofermentative lactic acid bacteria and Lactobacillus buchneri is a heterofermentative lactic acid bacteria, the application of these two kinds of lactic acid bacteria in silage has been widely researched and practically verified, with mature technical system and application experience. A lot of literature on inoculation of two types of lactic acid bacteria in pasture was also referred to, so Lactobacillus plantarum and Lactobacillus buchneri were selected for the study.

Mechanism of action:

Lactobacillus plantarum: Lactobacillus plantarum mainly carries out homofermentation, i.e. the fermentation of sugars to produce lactic acid. This enables Lactobacillus plantarum to rapidly reduce the pH value of the feed at the beginning of silage, inhibit the growth of harmful microorganisms and ensure the fermentation quality of the silage. Lactobacillus plantarum, as a silage fermentation promoter, can significantly increase the basal number of lactic acid bacteria at the early stage of silage, thus accelerating the reproduction rate of lactic acid bacteria and producing more lactic acid. By lowering the pH value, Lactobacillus plantarum can effectively inhibit the growth of harmful microorganisms such as molds and yeasts, and reduce the nutrient loss of silage during storage.

Lactobacillus buchneri: Lactobacillus buchneri belongs to the heterofermentative lactic acid bacteria, which is able to anaerobically decompose glucose and other saccharides during the silage process, and produce not only lactic acid, but also acetic acid, ethanol and 1,2-propanediol. Acetic acid produced by Lactobacillus buchneri is the second highest concentration of acid in the silage process, and has a significant antibacterial effect. Acetic acid can inhibit the reproduction of yeast and other undesirable bacteria, improve the aerobic stability of silage, and reduce the secondary fermentation and spoilage of silage.

Point 4: The introduction section needs to highlight more clearly the innovation and contribution of this study to the existing body of knowledge, as well as its potential impact on the livestock and feed processing industrie.

Response 4: We fully appreciate your suggestion. We revised that section in the paper. (PDF:62-68, 116-118)

Point 5: Authors need to add time and location of silage fermentation and location of fistula sheep to the material approach.

Response 5: We fully appreciate your suggestion. We revised that section in the paper. (PDF:144-146, 207-210)

Point 6: The results section Figure 3.5A is stretched out of shape and the authors need to correct it by adjusting the size and style. Also, the authors need to label the results for significance of differences.

Response 6: We fully appreciate your suggestion. The time required for a 2 °C difference between the silage and ambient temperatures was used to assess the aerobic stability of the silage. Figure 1 is just the temperature change of the additive on SP silage exposed to aerobic conditions. Therefore, there is no need to label the significance of the differences in Figure 1. We only analyzed significant differences in aerobic stability for different additive treatments. We revised that section in the paper. (PDF:345)

Point 7: For observed experimental phenomena, the discussion section should go deeper into the biological mechanisms behind them. The authors may cite relevant literature to explain how lactic acid bacteria inhibit the growth of harmful microorganisms by lowering pH through the production of lactic acid, or discuss how lactic acid bacteria affect the degradation of plant cell walls, which in turn affects the in vitro digestibility of feed.

Response 7: We fully appreciate your suggestion. We revised that section in the paper. (PDF:376-384, 532-537)

Point 8: The discussion section should give more consideration to the practical application value of the research results. The potential impact of the combined addition of Lactobacillus plantarum and Lactobacillus buchneri to Silphium perfoliatum L. silage on the productivity and economic benefits of animal husbandry could be explored. Also, problems and solutions that may be encountered in practical applications can be proposed.

Response 8: We fully appreciate your suggestion. We revised that section in the paper. (PDF:500-506, 509-516)

Thank you again for your suggestions and hope to learn more from you.

Round 2

Reviewer 1 Report

Comments and Suggestions for Authors

The authors have worked on the corrections and provided answers to my questions. Therefore, I decide to accept.